# Exploring the risks of over-reliance on AI in diagnostic pathology. What lessons can be learned to support the training of young pathologists?

Yaëlle Bellahsen-Harrar[1,2]⊙, Mélanie Lubrano[34]⊙, Charles Lépine[5,6,7],
Aurélie Beaufrère[3,8], Claire Bocciarelli[9], Anaïs Brunet[10,11], Elise Decroix[12],
Franck Neil El-Sissy[13], Bettina Fabiani[4], Aurélien Morini[13,14], Cyprien Tilmant[15],
Thomas Walter[3,16,17]⊙, Cécile Badoual[1,2,*]⊙

**1** Department of Pathology, Hôpital Européen Georges-Pompidou, APHP, Paris, France, **2** Université Paris Cité, Paris, France, **3** Centre for Computational Biology (CBIO), Mines Paris, PSL University, Paris, France, **4** Tribun Health, Paris, France, **5** Department of Pathology, CHU Nantes, Nantes, France, **6** UMR 1302/EMR6001, INCIT, Inserm, Nantes, France, **7** Nantes Université, Nantes, France, **8** Department of Pathology, Hôpital Beaujon, APHP, Clichy, France, **9** Department of Pathology, CHRU Brest, Brest, France, **10** Department of Pathology, Hôpital Henri Mondor, APHP, Créteil, France, **11** Université Paris Est Créteil, Créteil, France, **12** Department of Pathology, IUCT-Oncopole, Toulouse, France, **13** Department of Pathology, Hôpital Lariboisière, APHP, Paris, France, **14** Department of Pathology, Grand Hôpital de l'Est Francilien, Jossigny, France, **15** Department of Pathology, Groupement des Hôpitaux de l'Institut Catholique de Lille, Lille, France, **16** Institut Curie, PSL Université, Paris, France, **17** U1331 INSERM « Computational Oncology », Paris, France

⊙ These authors contributed equally to this work.
* cecile.badoual@aphp.fr

## Abstract

The integration of Artificial Intelligence (AI) algorithms into pathology practice presents both opportunities and challenges. Although it can improve accuracy and inter-rater reliability, it is not infallible and can produce erroneous diagnoses, hence the need for pathologists to always check predictions. This critical judgment is particularly important when algorithm errors could lead to high-impact negative clinical outcomes, such as missing an invasive carcinoma. However, the influence of AI tools on pathologists' decision-making is not well explored. This study aims to evaluate the impact of a previously developed AI tool on the diagnostic accuracy and inter-rater reliability among pathologists, while assessing whether pathologists maintain independent judgment of AI predictions. Eight pathologists from different hospitals and with varying levels of experience, participated in the study. Each of them reviewed 115 slides of laryngeal biopsies, including benign epithelium, low-grade and high-grade dysplasia, and invasive squamous carcinomas. The study compared diagnostic outcomes with and without AI assistance. The reference labels were established by an expert's double-blind review. Results show that assisted pathologists had a higher accuracy for high-grade dysplasia, invasive carcinoma and improved inter-rater reliability. However, cases of over-reliance on AI have been observed, resulting in the omission

**Data availability statement:** All relevant anonymized data are available in the paper and its Supporting Information files.

**Funding:** This work was supported by funding awarded to ML by a CIFRE PhD Fellowship founded by TRIBUN HEALTH, Paris, France and ANRT (CIFRE 2019/1905). Furthermore, this work was supported by the Ministère de l'Enseignement supérieur, de la Recherche et de l'Innovation under management of Agence Nationale de la Recherche as part of the "Investissements d'avenir" program, reference ANR-19-P3IA-0001 (PRAIRIE 3IA Institute). The funders had no role in study design, data collection and analysis, decision to publish or preparation of the manuscript. The confidence score method is detailed in a previous article cited in the references. The method related to the confidence index is the subject of a priority French patent application, FR 2213863, filed on December 19, 2022, in the names of AP-HP, Tribun Health, École des Mines, and Armines. As a reference, the application can be described as follows: Badoual C., Bellahsen Y., Lubrano M., Walter T., "Method for assisting in the classification of a patient's histological section, associated device," patent application FR 2213863 filed on 12/19/2022.

**Competing interests:** The authors have declared that no competing interests exist.

of correctly diagnosed invasive carcinomas during the unassisted examination. The false predictions on these carcinoma slides were labeled with a low confidence score, which was not considered by the less experienced pathologists, showing the risk that they would follow the AI prediction without enough critical judgment or expertise. Our study emphasizes the potential over-reliance of pathologists on AI models and the potential harmful consequences, even with the advancement of powerful algorithms. The integration of confidence scores and the education of pathologists to use this tool could help to optimize the safe integration of AI into pathology practice.

## Introduction

As digital pathology platforms become more widespread, the integration of Artificial Intelligence (AI) in diagnostic pathology is beginning to transform traditional workflows, promising to be a significant milestone in pathology practice. However, little attention has been paid to the impact of diagnostic tools on pathologists' decision-making. Although AI globally enhances diagnostic accuracy, it could also lead to misdiagnoses if pathologists uncritically follow AI recommendations. Most recent publications have focused on AI's ability to improve raw diagnostic performance but have not explored the potential clinical negative impacts of misleading predictions [1,2]. For AI tools to gain acceptance within the medical community, pathologists must trust these systems while retaining the ability to critically assess AI-generated predictions, particularly for high-impact diagnoses [3]. However, to our knowledge, no studies have thoroughly examined whether pathologists can maintain independent judgment when faced with AI suggestions, or their ability to detect false predictions.

The objectives of this study were to provide an in-depth evaluation of a diagnostic aid that distinguishes normal tissue, premalignant lesions and invasive carcinomas in the head and neck for pathologists with different levels of expertise. Head and neck squamous cell carcinomas are a significant global health concern, ranking sixth worldwide in both incidence and mortality rates [4]. These cancers are notoriously associated with poor prognosis and high morbidity in the laryngeal and pharyngeal regions. The diagnosis of invasive carcinomas and their pre-cancerous lesions, called squamous dysplasias, is a particularly challenging field since the diagnostic solely relies on morphological analysis with no immunostain to help the distinction between lesions [5]. Moreover, there is a shortage of pathology experts and a mediocre inter-rater agreement [6]. Misdiagnosis of high-grade dysplasias and invasive carcinomas have a high-impact negative outcome. Notably, a higher diagnostic delay of invasive carcinoma is associated with a worse overall survival [7,8]. The confusion between low-grade and high-grade dysplasias can also impair treatment decisions and patient follow-up [9].

In this work, we wanted to go beyond crude measures of performance and assess the pathologist's ability to judge potentially erroneous predictions, and the extent to which they can lead to an erroneous diagnosis. We analyzed misleading predictions in depth and discussed ways of reducing those with the most detrimental clinical consequences, taking account of pathologists' level of training and expertise.

## Materials and methods

Our study was approved by the Ethics Committee of Assistance Publique – Hôpitaux de Paris Centre (CERAPHP. Centre – Institutional Review Board registration #00011928). All the patients were informed by a notification letter of the study and the possibility to refuse the use of their medical data, in line with current legislation.

### Deep learning model and confidence score measurement

Based on the widely used Attention-MIL architecture [10], we developed, trained, and validated a model for automatic grading of head and neck squamous lesions in a previous work [11]. To develop this model, patients' data were accessed for research purposes since January 5, 2022. For each slide, two outputs were generated: the predicted lesion, ranging from non-dysplastic, low-grade dysplasia, high-grade dysplasia, to carcinoma, following the WHO classification system [5], and an associated confidence score. The confidence score is specifically designed to measure the model's level of confidence for lesions on the same spectrum: it measures the extent to which the model hesitated with the second most probable (adjacent) class, as described in a previous work [12]. Shortly, it is obtained from the risk estimation vector outputted by the last layer of the network. A SoftMax is applied to the inverted risk (- risk vector) to convert it into probabilities. The confidence score is then computed by measuring the difference between the two highest probabilities, thus assessing how much the model hesitates between the two most probable classes. Details on the computing methods are available in S1 File.

The model was trained using a monocentric dataset of 1949 digitized haematoxylin, eosin and saffron stained slides obtained from 456 patients who underwent either biopsies or surgical resection at Hôpital Européen Georges Pompidou (AP-HP, Paris, France). Each slide was associated with one class based on the most severe lesion present in the sample. The slides were digitized at 20X magnification using a Hamamatsu NanoZoomer® s360 scanner, resulting in a pixel resolution of 0.45 µm. To properly evaluate the model's performance, an independent subsample of 115 biopsies from 101 patients was used as the test set. The classes for these slides were determined using a dual-blind review by two pathologists with expertise in head and neck squamous lesions, followed by a consensus meeting to thoroughly discuss any slides on which they disagreed. This dataset was used to evaluate the performance of the AI model and to assess the effectiveness of AI-assisted reviews in terms of reproducibility between pathologists and their diagnostic performances. Finally, the model was validated on an external dataset from another center (Hôpital Tenon, AP-HP, Paris, France) including 87 slides from 67 patients. Details about the datasets are shown in S1 Table.

### Pathologist panel and trial design

The pathologist panel and trial design are presented in Fig 1. The panel consisted of eight pathologists coming from different hospitals in France, with varying experience levels and practice backgrounds: two newly board-certified, and five board-certified experienced pathologists, three of them specialized in head and neck pathology. The two expert pathologists who labeled the slides for the algorithm validation were not included in this panel.

All panel members were tasked with reviewing the slides from the reference standard test set with and without AI-Assistance. Residents and non-specialized pathologists were provided with the references of the WHO grading system beforehand to update their knowledge. The study was designed as a randomized crossover trial, where each participant was randomly assigned to start with either the AI-assisted review or the unassisted review, following protocols used in other studies [13–15]. The pathologists independently examined the subset of 115 slides reviewed by the experts and assigned a diagnosis to each slide without external input, in an uninterrupted session. A mandatory washout period of at least two weeks was imposed between the two examinations to avoid potential carryover effects.

### Digital platform and review process

The reviews were conducted using a web-based viewer allowing for simultaneous visualization of the slides, the model's prediction, and the confidence score. A tutorial by videoconference and a written user guide were provided to the

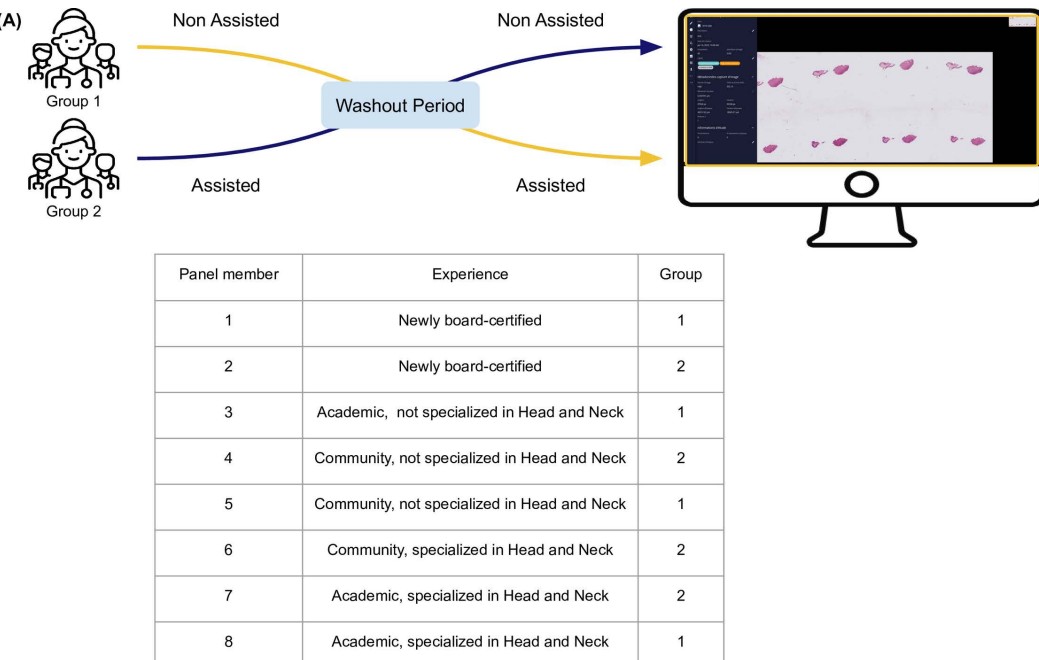

| Panel member | Experience | Group |
|---|---|---|
| 1 | Newly board-certified | 1 |
| 2 | Newly board-certified | 2 |
| 3 | Academic, not specialized in Head and Neck | 1 |
| 4 | Community, not specialized in Head and Neck | 2 |
| 5 | Community, not specialized in Head and Neck | 1 |
| 6 | Community, specialized in Head and Neck | 2 |
| 7 | Academic, specialized in Head and Neck | 2 |
| 8 | Academic, specialized in Head and Neck | 1 |

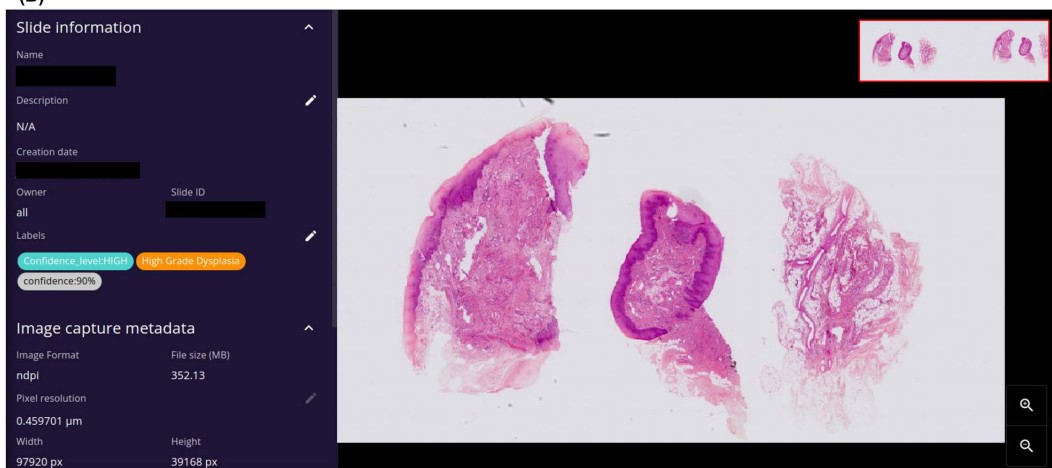

**Fig 1. Assisted review on the digital platform. (A)** Review protocol. The pathologists began either with assisted or non-assisted review and switched after a washout period of at least two weeks. **(B)** A whole slide image viewer. displaying for each case the virtual slide, the model's prediction, the confidence score expressed as a percentage, a categorization of the confidence (high or low) and a heatmap highlighting regions of the slide that contributed to the prediction.

participants. The assisted and unassisted examinations were carried out on the same platform, but the names of the slides were changed between the two examinations to ensure blinding. During the unassisted review, the pathologists had access only to the slides and were blind to any other information related to the case. For the assisted reviews, they were provided with the model's prediction, the confidence score expressed as a percentage, a categorization of the confidence (high or low, following the threshold established beforehand) and a heatmap that could be toggled on and off, highlighting regions of the slide that contributed to the prediction, as shown in Fig 2. For each slide, the evaluators

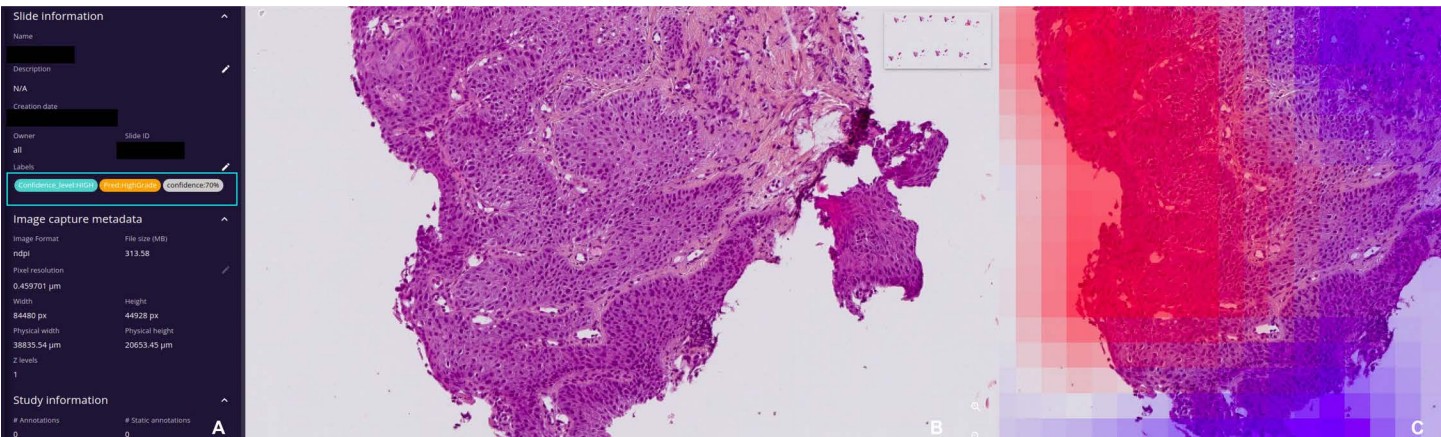

**Fig 2. Digital web-based review platform. (A)** Slide information. Framed in blue: model's prediction, confidence score (%) and confidence level (low or high). **(B)** Virtual slide. **(C)** Overlaid heatmap showing in red the tiles that drove the model's attention and in blue the tiles with a low attention score.

were asked to fill in a table with their diagnosis, with the names of the slides pre-filled in the order in which they appeared on the platform.

## Statistical analysis

Once all panel members had completed the assisted and unassisted examinations, their diagnoses were compared with the slide's labels. Cohen's kappa with linear weights was used as the primary metric to measure the reproducibility, to compare it with the other studies published in the literature [6]. Other standard classification metrics (accuracy, sensitivity, specificity, negative and positive predictive values) were computed per class in a one-versus-rest manner. Confidence intervals of the AI algorithm model were computed using 10000 bootstraps. Statistical differences of the metrics between the AI-Assisted and the unassisted reviews were assessed with a paired t-test. To account for a possible bias in the reference standard of the internal test set, the pairwise agreement between all panel members was calculated individually for both tests. All statistical analyses were performed using python (v3.6.9), pandas (v1.1.5), scikit learn (v1.2.0) and SciPy (v1.6.0).

## Results

The aims of this study were to determine the added value of our AI tool on the pathologist's performances and to assess if they could keep an independent judgment in front of the predictions. Key issues included the detection of high-grade dysplasia and the identification of invasive carcinoma, which are the diagnoses with the most impactful clinical consequences.

### Agreement between pathologists with and without AI assistance

Agreement analysis is presented in Fig 3. The results show that AI-Assistance significantly improved inter-rater agreement, as indicated by the reduced range of kappa values (unassisted review: linear kappa's range from 0.576 to 0.742; assisted review: linear kappa's range from 0.698 to 0.767). The mean linear kappa of the unassisted review was like the standalone model (0.675, 95%CI [0.579–0.765]) whereas the assisted review outperformed the AI (mean linear kappa: 0.73, 95%CI [0.711–0.748], p < 0.001). When considering pairwise agreement within the panel without taking the reference standard labels into account, the mean linear kappa was 0.616 (95%CI [0.597–0.637]) in the unassisted review and 0.736 (95%CI [0.721–0.752]) in the assisted review (p < 0.001). These results show that AI-Assistance led to increased consistency in grading among the pathologists.

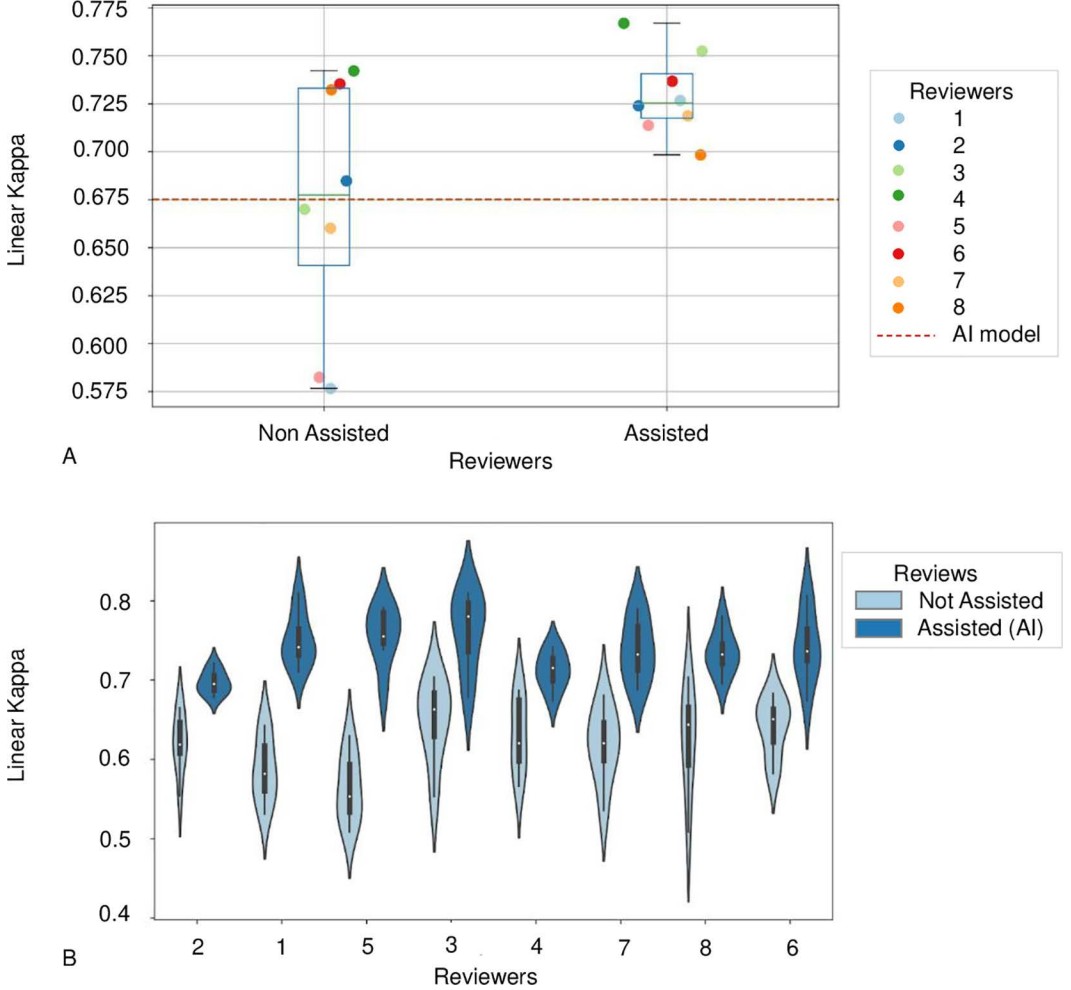

**Fig 3. Agreement between pathologists with and without AI-assistance. (A)** Overall linear kappa. These plots show the linear kappa values for each pathologist with and without the AI-Assistance, compared to the standalone AI-model. The assisted review improved the inter-rater agreement and drastically reduced the kappa's range between pathologists. **(B)** Pairwise agreement. This figure illustrates the increase of the inter-rater agreement without considering the reference standard labels (linear kappa 0.734 assisted versus 0.619 non-assisted).

## Performance improvements in diagnosing high grade dysplasia and invasive carcinoma

Detailed pathologists' performances per diagnostic class are shown in Table 1. Notably, performances showed significant improvements in diagnosing high-grade dysplasia and carcinoma with AI assistance, outperforming the standalone model. Assisted pathologists obtained a higher accuracy for high-grade dysplasia (0.774, 95%CI [0.696–0.844] vs 0.803, 95%CI [0.795–0.812], p = 0.003), invasive carcinoma (0.896, 95%CI [0.835–0.948] vs 0.927, 95%CI [0.911–0.943], p = 0.004), and reliability (linear kappas: 0.675, 95%CI [0.579–0.765] vs 0.73, 95%CI [0.711–0.748], p < 0.001). These results show that the AI tool assisted pathologists in identifying these harmful lesions with the highest therapeutic impact, and the combination of the pathologist's expertise and the AI analysis proved to be complementary and more powerful when used together. Moreover, specificity improved significantly for low-grade dysplasia, indicating better discrimination of this subtle lesion (unassisted pathologists: 0.823, 95%CI [0.788–0.858] versus assisted pathologists: 0.867, 95%CI [0.854–0.880], p = 0.022).

**Table 1. Classification metrics per type of lesion.**

| Metric | Status | Type of lesion | | | |
|---|---|---|---|---|---|
| | | Non-dysplastic | Low-grade dysplasia | High-grade dysplasia | Invasive carcinoma |
| Accuracy | Non assisted Assisted (P-value*) | 0.837 [0.813-0.861] 0.866 [0.853-0.880] (0.053) | 0.751 [0.722-0.781] 0.786 [0.770-0.802] (0.068) | 0.788 [0.765-0.811] 0.803 [0.795-0.812] (0.119) | **0.902** **[0.884-0.920]** **0.927** **[0.911-0.943]** **(0.041)** |
| | Standalone model Assisted (P-value*) | 0.861 [0.791-0.922] 0.866 [0.853-0.880] (0.224) | 0.757 [0.670-0.835] 0.786 [0.770-0.802] (0.986) | **0.774** **[0.696-0.844]** **0.803** **[0.795-0.812]** **(0.003)** | **0.896** **[0.835-0.948]** **0.927** **[0.911-0.943]** **(0.004)** |
| Negative Predictive Value (NPV) | Non assisted Assisted (P-value*) | **0.882** **[0.857-0.906]** **0.917** **[0.893-0.941]** **(0.022)** | 0.864 [0.846-0.881] 0.869 [0.854-0.884] (0.338) | 0.867 [0.843-0.891] 0.871 [0.849-0.893] (0.384) | **0.916** **[0.889-0.942]** **0.944** **[0.922-0.965]** **(0.029)** |
| | Standalone model Assisted (P-value*) | 0.911 [0.851-0.966] 0.917 [0.893-0.941] (0.193) | 0.882 [0.802-0.95] 0.869 [0.854-0.884] (0.942) | **0.819** **[0.736-0.892]** **0.871** **[0.849-0.893]** **(0.000)** | **0.921** **[0.857-0.973]** **0.944** **[0.922-0.965]** **(0.048)** |
| Positive Predictive Value (PPV) (Precision) | Non assisted Assisted (P-value*) | 0.657 [0.580-0.734] 0.698 [0.660-0.736] (0.206) | 0.382 [0.314-0.451] 0.438 [0.393-0.482] (0.163) | 0.581 [0.534-0.627] 0.613 [0.591-0.634] (0.075) | 0.880 [0.856-0.903] 0.899 [0.870-0.927] (0.128) |
| | Standalone model Assisted (P-value*) | 0.680 [0.480-0.864] 0.698 [0.660-0.736] (0.406) | 0.400 [0.219-0.572] 0.438 [0.393-0.482] (0.071) | **0.571** **[0.333-0.800]** **0.613** **[0.591-0.634]** **(0.001)** | **0.846** **[0.722-0.946]** **0.899** **[0.870-0.927] (0.003)** |
| Sensitivity (Recall) | Non assisted Assisted (P-value*) | **0.550** **[0.440-0.660]** **0.695** **[0.600-0.790]** **(0.024)** | 0.449 [0.367-0.530] 0.443 [0.370-0.516] (0.546) | 0.608 [0.518-0.697] 0.612 [0.530-0.695] (0.464) | **0.827** **[0.765-0.889]** **0.888** **[0.842-0.933]** **(0.029)** |
| | Standalone model Assisted (P-value*) | 0.680 [0.517-0.862] 0.695 [0.600-0.790] (0.193) | 0.545 [0.323-0.750] 0.443 [0.370-0.516] (0.527) | 0.414 [0.240-0.593] 0.612 [0.530-0.695] (0.969) | **0.846** **[0.727-0.946]** **0.888** **[0.842-0.933]** **(0.004)** |
| Specificity | Non assisted Assisted (P-value*) | 0.917 [0.891-0.942] 0.914 [0.892-0.936] (0.583) | **0.823** **[0.788-0.858]** **0.867** **[0.854-0.880]** **(0.022)** | 0.849 [0.815-0.883] 0.868 [0.843-0.892] (0.073) | 0.941 [0.926-0.955] 0.947 [0.930-0.965] (0.306) |
| | Standalone model Assisted (P-value*) | 0.911 [0.849-0.965] 0.914 [0.892-0.936] (0.383) | 0.806 [0.725-0.882] 0.867 [0.854-0.880] (0.071) | **0.895** **[0.829-0.956]** **0.868** **[0.843-0.892]** **(0.000)** | **0.921** **[0.855-0.974]** **0.947** **[0.930-0.965]** **(0.011)** |

* one-sided paired t tests. Results with statistical significance are in bold.

## Disagreements between the AI and the pathologists during the assisted review

On average, the pathologists disagreed with the AI predictions in 32% of cases, with no significant difference between the panel members. For around 30% of these cases, the pathologists corrected AI misdiagnoses. Notably, the model mistakenly classified (only) 3 high grade dysplasia as carcinoma. The 3 cases were all corrected by the pathologists. Two of these cases had a high confidence score. The model did not make any other misclassification on the carcinoma class.

For the slides where the pathologists were wrong and the AI prediction right, the main source of discordance was between normal epithelium and low-grade dysplasia, which are very subjective differences by essence and with a low clinical impact.

Examples of disagreements between normal epithelium and low-grade dysplasia are shown in Fig 4.

## Analysis of high-impact clinical mistakes induced by the AI

Aggregating the 8 assisted reviews of the 115 cases, a total of 920 reviews including 312 carcinoma slides were performed by pathologists with the help of AI. Among these assisted reviews, diagnosis was missed 35 times, including 19 slides wrongly diagnosed by both the model and the pathologist. These induced errors concentrated on only 6 cases, which all had low confidence scores and showed artifacts, tangent cuts, or presentations such as an invasive carcinoma with a normal epithelial surface above it. These 6 cases are shown in Fig 5. For 4 of them (5 slides reviewed), the pathologists were able to detect the carcinoma and made the correct diagnosis during their unassisted review. Interestingly, the false diagnostics made by specialized pathologists had a lesser impact compared to their non-specialized colleagues. For example, reviewer 6 (a community pathologist specialized in head and neck) accepted the AI prediction of high-grade dysplasia, which has a less severe outcome than completely missing an invasive lesion or mischaracterizing it as low-grade. Reviewer 7 (an academic pathologist specialized in head and neck), while unable to make the correct diagnosis, rejected the AI's low-grade dysplasia prediction and chose high-grade dysplasia instead. In contrast, less experienced pathologists (reviewers 1- recently board-certified, and 5- community pathologist not specialized in head and neck) followed the AI

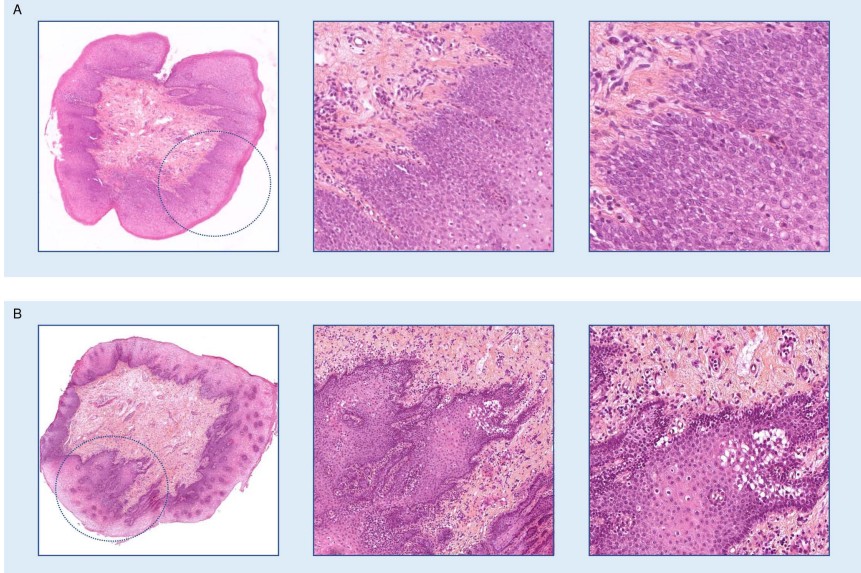

**Fig 4. Examples of disagreements between AI model and pathologists of low clinical impact.** Magnification from left to right: x2, x10, x20. A. slide_1, a normal epithelium wrongly diagnosed as low-grade dysplasia, which had a tangent inclusion potentially difficult to interpret. B. slide_2, a normal epithelium wrongly diagnosed as low-grade dysplasia, with a tangent cut of the basal layers which can be misleading.

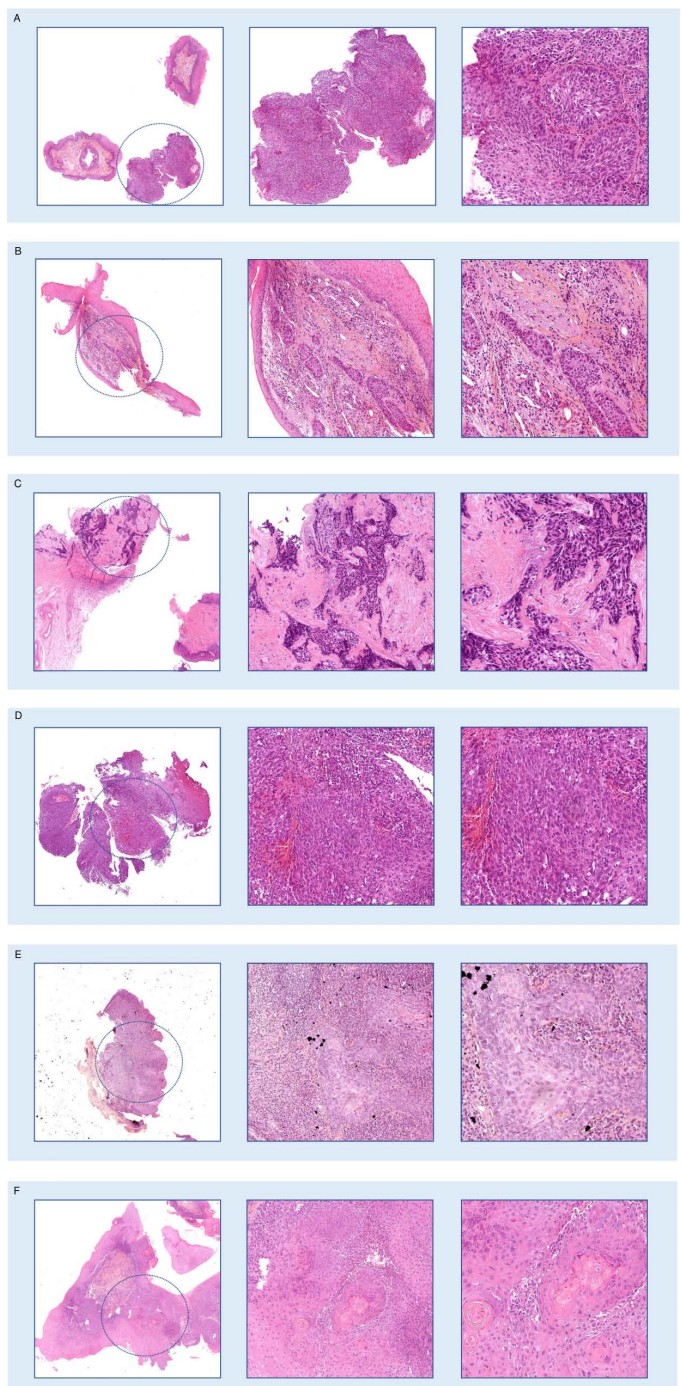

**Fig 5. Wrongly predicted carcinomas missed by the pathologists during the assisted reviews.** Magnification from left to right: x2, x10, x20. **A.** Slide_3: sometimes correctly diagnosed during the non assisted review, this slide shows tangent cuts making the prediction more difficult for the AI model. **B.** Slide_4: correctly diagnosed during the non assisted review, this slide shows invasive carcinoma under a normal epithelium, which is an unusual presentation. **C.** Slide_5: correctly diagnosed during the non assisted review, this slide shows crushing artifacts, which might impair the prediction. **D.** Slide_6: missed during the non assisted review, this slide shows tangent cuts difficult to interpret. **E.** Slide_7: missed during the non assisted review, this slide also shows tangent cuts. **F.** Slide_8: sometimes correctly diagnosed during the non assisted review, this slide shows tangent cuts.

predictions, which were significantly inaccurate (low-grade dysplasia or non-dysplastic epithelium). Detailed results are shown in S2 Table.

For the 15 other slides, which were correctly predicted as carcinoma by the model, most had a high confidence score (12/15), however the pathologists always considered them as high-grade dysplasia.

### Impact of the confidence score on the pathologists' diagnostic choices

When considering the model's confidence scores (as shown in Fig 6), the results indicate that on high confidence predictions, reproducibility was significantly higher and with a reduced distribution of kappa values (high confidence predictions: linear kappa 0.809, 95%CI [0.784–0.834] for assisted review, versus 0.731, 95%CI [0.681–0.781] unassisted, p = 0.018). There was no difference in the kappa values between assisted and unassisted reviews when the confidence score was below the threshold (low confidence predictions: linear kappa 0.533, 95%CI [0.483–0.583] for assisted review, versus 0.522, 95%CI [0.459–0.586] unassisted, p = 0.342).

## Discussion

The first papers published on AI applied to pathology diagnostics revolutionized the field of computational pathology, making the first step towards integration of AI models in practice [16,17]. These works focused on the raw performances of the standalone models but didn't explore the impact of these tools when integrated into a pathologist's workflow. More recently, several studies have shown the benefits of AI models for improved diagnostic accuracy and reproducibility among pathologists, leading to what could be called an "augmented pathology" [13,18,19]. However, most of them didn't analyze the capacity of the pathologists to keep an independent judgment in front of false predictions, and the clinical consequences of these induced mistakes. Indeed, pathologists may become overly influenced by the model and less critical of its outputs, leading to problematic outcomes.

This potential wrong influence could be particularly damaging when pathologists must discern between a wide and subtle range of lesions from benign, premalignant, and invasive. To our knowledge, previous works in the field didn't truly pay attention to the full complexity of this kind of setting since they focused on classification between different cancer subtypes or carcinoma grading. The work of Ba et al. [18] on augmented pathology in gastric lesions encompassed benign and malignant lesions but clustered for the statiscial analyses malignant and possibly malignant lesions in one group, benign and possibly benign lesions in another, limiting the final output to a binary choice not reflecting real-life practice. Even if our dataset focuses on a specific anatomic area (the larynx and hypopharynx), it differentiates between benign epithelium, low-grade, high-grade dysplasias and invasive carcinomas following the WHO grading system used in routine practice.

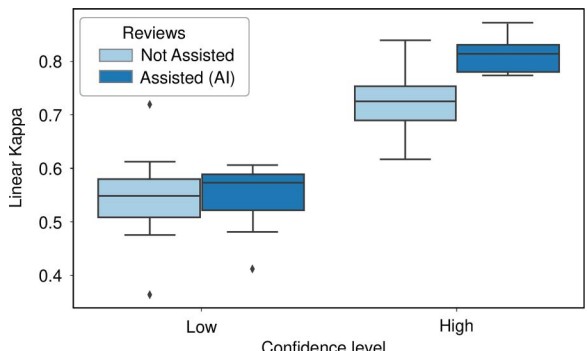

**Fig 6. Pathologists' performances depending on the confidence level.** The model's confidence score guided the pathologists and improved the inter-rater agreement, with higher linear kappa on confident predictions (high confidence predictions: linear kappa 0.8 assisted versus 0.73 non-assisted, p < 0.001).

In line with a previous work of Raciti et al. on the binary diagnostic task of prostate cancer detection [20], we show that the benefice/risk balance is in favor of the use of AI even on expert fields such as squamous head and neck lesions, showing that AI improves overall performances while leading to exceptional misdiagnosis. However, to date, these recent works didn't present the rare, but inevitable, AI-induced wrong diagnostics and ways to reduce them, nor evaluated their potentially harmful clinical consequences. In our work, the worst-impact errors induced by the model concentrated on particular settings which were filtered out as low-confident predictions by our scoring method. Nevertheless, our results show that the non-specialized pathologists didn't consider the confidence score in their decision, relying on the model's prediction in these cases, even if they were able to make the correct diagnosis without the AI assistance. Our results indicate that a written protocol and oral explanation about model usage are insufficient for less experienced pathologists. To help them use the model more efficiently, low-confidence predictions should be more stressed out, for example with a pop-up alert. This could be a simple and efficient way to address the higher reliability of non-specialized and in-training pathologists.

In this work, we could not directly demonstrate that training younger and non-specialized pathologists on the panel would enhance their ability to use AI effectively and improve diagnostic accuracy. To establish this, a repeated evaluation of the slides would have been necessary to observe whether their performance improves through iterative learning from errors. However, because our dataset is limited, particularly concerning misinterpreted cases, there is the concern that pathologists may remember their previous incorrect diagnoses and provide correct answers through memory recall rather than genuine improvement in diagnostic skills, in the same way an AI model can be overfit on limited datasets. Such investigations would need a longitudinal study, tracking pathologists from the initial stages of their training through their early years of board-certified practice.

In radiology, a field with substantial AI integration in daily practice, previous studies have indicated that younger radiologists are particularly prone to automation bias, resulting in an over-reliance on AI predictions [21–23]. A study by Mehrizi et al. [24] suggested that even increased explainability does not mitigate this issue. Our results show same issues apply to AI-assisted pathology. Notably, in cases where the prediction led to incorrect diagnoses, the more experienced pathologists considered the AI's input but did not follow it uncritically. In contrast, recently board-certified and non-specialized pathologists were more likely to rely on the model's predictions. This may show that tendency to automation bias in younger practitioners might be related to the "figure of authority" described in cognitive development science, here consisting in a mathematical algorithm which operating mode is not well understood, and thus not really questionable when lacking self-confidence [25]. These considerations underscore the need for alternative approaches to assist pathologists in using AI effectively.

To address these challenges, radiology societies in North America have introduced structured educational programs, such as the National Imaging Informatics Course – Radiology and the Radiological Society of North America Imaging AI Certificate [26]. Notably, these courses teach how AI models work and their ability to induce biased judgments. Some pathology societies are now following a similar path, advocating for AI-focused training that encompasses the recognition of AI limitations and the risks associated with its use [27]. In France, the integration of a dedicated AI mandatory course is quite recent (2023), and it has yet to include a specific course on AI-induced bias and risks of over-reliance.

In conclusion, our findings show the potential for AI assistance to lead to clinically impactful diagnostic errors if not applied cautiously. These findings are consistent with previous observations made in AI-assisted radiology. Our results highlight the importance of early integration of AI limitation awareness and the need for follow-up of young pathologists, who are the most vulnerable to automation bias and over-reliance. Incorporating confidence measurements into the decision-making process could help address this issue and should be encouraged. Assessing the impact of confidence scores on pathologist training would require a dedicated, long-term study.

## Supporting information

**S1 File. Whole slide images classification code.**
(ZIP)

**S1 Table. Datasets description.** This table summarizes the data used for the model training, internal validation and external validation. Training and internal validation were performed on a total of 2064 slides from Hôpital Européen Georges Pompidou, and external validation was made on a set of 87 slides from Hôpital Tenon (APHP, Paris, France). (XLSX)

**S2 Table. Detailed results of induced missed carcinomas.**
(XLSX)

## Author contributions

**Conceptualization:** Yaëlle Bellahsen-Harrar, Mélanie Lubrano, Thomas Walter, Cécile Badoual.

**Data curation:** Yaëlle Bellahsen-Harrar, Mélanie Lubrano.

**Formal analysis:** Yaëlle Bellahsen-Harrar, Mélanie Lubrano, Charles Lépine, Aurélie Beaufrère, Claire Bocciarelli, Anaïs Brunet, Elise Decroix, Franck Neil El-Sissy, Bettina Fabiani, Aurélien Morini, Cyprien Tilmant.

**Funding acquisition:** Mélanie Lubrano, Thomas Walter.

**Investigation:** Yaëlle Bellahsen-Harrar, Mélanie Lubrano, Cécile Badoual.

**Methodology:** Yaëlle Bellahsen-Harrar, Mélanie Lubrano, Cécile Badoual.

**Project administration:** Thomas Walter, Cécile Badoual.

**Resources:** Cécile Badoual.

**Software:** Mélanie Lubrano.

**Supervision:** Thomas Walter, Cécile Badoual.

**Validation:** Yaëlle Bellahsen-Harrar, Mélanie Lubrano, Cécile Badoual.

**Visualization:** Yaëlle Bellahsen-Harrar, Mélanie Lubrano, Cécile Badoual.

**Writing – original draft:** Yaëlle Bellahsen-Harrar, Cécile Badoual.

**Writing – review & editing:** Yaëlle Bellahsen-Harrar, Mélanie Lubrano, Charles Lépine, Aurélie Beaufrère, Claire Bocciarelli, Anaïs Brunet, Elise Decroix, Franck Neil El-Sissy, Bettina Fabiani, Aurélien Morini, Cyprien Tilmant, Thomas Walter, Cécile Badoual.

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
