## [Decision Letter · Decision Letter 0]

23 Dec 2024

PONE-D-24-53739AI in diagnostic pathology: exploring the risks of over-reliance and its clinical consequences. What lessons can be learned to support the training of young pathologists?PLOS ONE

Dear Dr. Bellahsen-Harrar,

Thank you for submitting your manuscript to PLOS ONE. After careful consideration, we feel that it has merit but does not fully meet PLOS ONE’s publication criteria as it currently stands. Therefore, we invite you to submit a revised version of the manuscript that addresses the points raised during the review process.

We look forward to receiving your revised manuscript.

Kind regards,

Mohammad Amin Fraiwan

Academic Editor

PLOS ONE

Journal Requirements: When submitting your revision, we need you to address these additional requirements. 1. Please ensure that your manuscript meets PLOS ONE's style requirements, including those for file naming. The PLOS ONE style templates can be found at https://journals.plos.org/plosone/s/file?id=wjVg/PLOSOne_formatting_sample_main_body.pdf and https://journals.plos.org/plosone/s/file?id=ba62/PLOSOne_formatting_sample_title_authors_affiliations.pdf 2. Please note that PLOS ONE has specific guidelines on code sharing for submissions in which author-generated code underpins the findings in the manuscript. In these cases, we expect all author-generated code to be made available without restrictions upon publication of the work. Please review our guidelines at https://journals.plos.org/plosone/s/materials-and-software-sharing#loc-sharing-code and ensure that your code is shared in a way that follows best practice and facilitates reproducibility and reuse. 3. We note that the grant information you provided in the ‘Funding Information’ and ‘Financial Disclosure’ sections do not match.  When you resubmit, please ensure that you provide the correct grant numbers for the awards you received for your study in the ‘Funding Information’ section. 4. Thank you for stating the following financial disclosure: "ML was supported by a CIFRE PhD Fellowship founded by TRIBUN HEALTH, Paris, France and ANRT (CIFRE 2019/1905). Furthermore, this work was supported by the Ministère de l’Enseignement supérieur, de la Recherche et de l’Innovation under management of Agence Nationale de la Recherche as part of the “Investissements d'avenir” program, reference ANR-19-P3IA-0001 (PRAIRIE 3IA Institute). " Please state what role the funders took in the study.  If the funders had no role, please state: ""The funders had no role in study design, data collection and analysis, decision to publish, or preparation of the manuscript.""  If this statement is not correct you must amend it as needed. Please include this amended Role of Funder statement in your cover letter; we will change the online submission form on your behalf. 5. Thank you for stating the following in the Acknowledgments Section of your manuscript: "ML was supported by a CIFRE PhD Fellowship founded by TRIBUN HEALTH, Paris, France and ANRT (CIFRE 2019/1905). Furthermore, this work was supported by the Ministère de l’Enseignement supérieur, de la Recherche et de l’Innovation under management of Agence Nationale de la Recherche as part of the “Investissements d'avenir” program, reference ANR-19-P3IA-0001 (PRAIRIE 3IA Institute). TW acknowledges financial support from ITMO Cancer (20CM107-00)." We note that you have provided funding information that is not currently declared in your Funding Statement. However, funding information should not appear in the Acknowledgments section or other areas of your manuscript. We will only publish funding information present in the Funding Statement section of the online submission form. Please remove any funding-related text from the manuscript and let us know how you would like to update your Funding Statement. Currently, your Funding Statement reads as follows: "ML was supported by a CIFRE PhD Fellowship founded by TRIBUN HEALTH, Paris, France and ANRT (CIFRE 2019/1905). Furthermore, this work was supported by the Ministère de l’Enseignement supérieur, de la Recherche et de l’Innovation under management of Agence Nationale de la Recherche as part of the “Investissements d'avenir” program, reference ANR-19-P3IA-0001 (PRAIRIE 3IA Institute). "  Please include your amended statements within your cover letter; we will change the online submission form on your behalf. 6. We note that you have indicated that there are restrictions to data sharing for this study. For studies involving human research participant data or other sensitive data, we encourage authors to share de-identified or anonymized data. However, when data cannot be publicly shared for ethical reasons, we allow authors to make their data sets available upon request. For information on unacceptable data access restrictions, please see http://journals.plos.org/plosone/s/data-availability#loc-unacceptable-data-access-restrictions.  Before we proceed with your manuscript, please address the following prompts: a) If there are ethical or legal restrictions on sharing a de-identified data set, please explain them in detail (e.g., data contain potentially identifying or sensitive patient information, data are owned by a third-party organization, etc.) and who has imposed them (e.g., a Research Ethics Committee or Institutional Review Board, etc.). Please also provide contact information for a data access committee, ethics committee, or other institutional body to which data requests may be sent. b) If there are no restrictions, please upload the minimal anonymized data set necessary to replicate your study findings to a stable, public repository and provide us with the relevant URLs, DOIs, or accession numbers. Please see http://www.bmj.com/content/340/bmj.c181.long for guidelines on how to de-identify and prepare clinical data for publication. For a list of recommended repositories, please see https://journals.plos.org/plosone/s/recommended-repositories. You also have the option of uploading the data as Supporting Information files, but we would recommend depositing data directly to a data repository if possible. Please update your Data Availability statement in the submission form accordingly. 7. When completing the data availability statement of the submission form, you indicated that you will make your data available on acceptance. We strongly recommend all authors decide on a data sharing plan before acceptance, as the process can be lengthy and hold up publication timelines. Please note that, though access restrictions are acceptable now, your entire data will need to be made freely accessible if your manuscript is accepted for publication. This policy applies to all data except where public deposition would breach compliance with the protocol approved by your research ethics board. If you are unable to adhere to our open data policy, please kindly revise your statement to explain your reasoning and we will seek the editor's input on an exemption. Please be assured that, once you have provided your new statement, the assessment of your exemption will not hold up the peer review process. 8. Please include your full ethics statement in the ‘Methods’ section of your manuscript file. In your statement, please include the full name of the IRB or ethics committee who approved or waived your study, as well as whether or not you obtained informed written or verbal consent. If consent was waived for your study, please include this information in your statement as well. 9. Please include captions for your Supporting Information files at the end of your manuscript, and update any in-text citations to match accordingly. Please see our Supporting Information guidelines for more information: http://journals.plos.org/plosone/s/supporting-information.

Reviewers' comments:

Reviewer's Responses to Questions

**Comments to the Author**

1. Is the manuscript technically sound, and do the data support the conclusions?

Reviewer #1: Yes

Reviewer #2: Yes

2. Has the statistical analysis been performed appropriately and rigorously? 

Reviewer #1: Yes

Reviewer #2: Yes

3. Have the authors made all data underlying the findings in their manuscript fully available?

Reviewer #1: Yes

Reviewer #2: No

4. Is the manuscript presented in an intelligible fashion and written in standard English?

Reviewer #1: Yes

Reviewer #2: Yes

5. Review Comments to the Author

Reviewer #1: Dear editor, dear authors,

This paper addresses an important and often overlooked topic. How to deal with drawbacks of AI systems and how to avoid overconfidence in AI-based predictions?

Based on a subset of 115 cases/slides of head and neck biopsies (squamous lesions), the authors show that, although AI models generally lead to better predictions and help mainly less experienced pathologists, there are instances in which AI models generate errors. Pathologists (in this paper) seem to accept these errors and seem not to look at confidence scores generated by AI models. Although the topic is important, I fear the authors overgeneralize some findings.

Therefore I have a few comments which I hope the authors can address:

1. The performance of a single AI model is shown, in a single region of France. How sure are the authors that the problem of overreliance is general and not specific to the setting? What about different AI models?

2. What is the background on use of the AI model? Is this a model validated and in clinical use in France? Is this model currently in use in diagnostics? You mention how it was trained and developed, but after development, a validation set was used before using the model in clinical practice?

3. Is this a model routinely used on a daily basis or a RUO model? Are people using this model also using other AI models? Have they been trained and informed on the use of confidence scores? Are they aware of the potential problems of the model?

4. Please include an experiment on how to avoid these errors. Please show if (or not) training of pathologists on potential errors (before use of the AI) can make a difference. Or maybe a pop-up or warning of low confidence can help. Anything that mitigates these problems.

Then some minor comments:

-I would specify from how many cases these 115 slides were. Were these 115 slides from 115 cases or 115 slides from less cases (if multiple slides per case)

-Although you mention the confidence score of the model was determined in a previous study, I think it would be good to repeat how this was done.

Best wishes

Reviewer #2: Overall, this a well written study that covers a very important aspect of using AI in practice. Numerous studies have shown that AI tools can improve the performance of pathologists, but only a few studies such as this one have also tried to measure the tendency of AI tools to sway the judgement of pathologists in the wrong direction, which is concerning since AI-based tools can also make mistakes.

I have one minor comment:

In the last paragraph of the paper, it was mentioned - "Our findings highlight the possibility of misleading diagnosis even by a powerful algorithm when not used with caution. In view of these results, the integration of the confidence measurement in the decision process could help reduce the number of errors, especially on critical diagnoses such as invasive carcinomas, and should be encouraged."

I agree that "integration of the confidence measurement in the decision process could help reduce the number of errors", but I would like to point out a confidence measure of 100% does not necessarily mean the model's prediction is correct. There are probably instances where the confidence measure can be very high or 100% (depending on the model) and still be wrong, and pathologists should be aware of this to avoid confusion and be inadvertently led to a wrong diagnosis.

6. PLOS authors have the option to publish the peer review history of their article (what does this mean? ). If published, this will include your full peer review and any attached files.

**Do you want your identity to be public for this peer review?** For information about this choice, including consent withdrawal, please see our Privacy Policy .

Reviewer #1: No

Reviewer #2: No

---

## [Author Response · Author response to Decision Letter 1]

19 Feb 2025

Concerning the journal requirements:

Thank you for your comment. We modified the manuscript and the file names so they could meet PLOS ONE’s style requirements.

2. Please note that PLOS ONE has specific guidelines on code sharing for submissions in which author-generated code underpins the findings in the manuscript. In these cases, we expect all author-generated code to be made available without restrictions upon publication of the work.

The code of the deep learning model can be accessed via a GitHub repository: https://github.com/MelanieLu.

The confidence score method is detailed in a previous article cited in the references. The method related to the confidence index is the subject of a priority French patent application, FR 2213863, filed on December 19, 2022, in the names of AP-HP, Tribun Health, École des Mines, and Armines. As a reference, the application can be described as follows: Badoual C., Bellahsen Y., Lubrano M., Walter T., "Method for assisting in the classification of a patient's histological section, associated device," patent application FR 2213863 filed on 12/19/2022.

And

"ML was supported by a CIFRE PhD Fellowship founded by TRIBUN HEALTH, Paris, France and ANRT (CIFRE 2019/1905). Furthermore, this work was supported by the Ministère de l’Enseignement supérieur, de la Recherche et de l’Innovation under management of Agence Nationale de la Recherche as part of the “Investissements d'avenir” program, reference ANR-19-P3IA-0001 (PRAIRIE 3IA Institute). " Please state what role the funders took in the study. If the funders had no role, please state: ""The funders had no role in study design, data collection and analysis, decision to publish, or preparation of the manuscript."" If this statement is not correct you must amend it as needed. Please include this amended Role of Funder statement in your cover letter; we will change the online submission form on your behalf.

Here is the final financial disclosure:

“This work was supported by a CIFRE PhD Fellowship founded by TRIBUN HEALTH, Paris, France and ANRT (CIFRE 2019/1905). Furthermore, this work was supported by the Ministère de l’Enseignement supérieur, de la Recherche et de l’Innovation under management of Agence Nationale de la Recherche as part of the “Investissements d'avenir” program, reference ANR-19-P3IA-0001 (PRAIRIE 3IA Institute). The funders had no role in study design, data collection and analysis, decision to publish or preparation of the manuscript.”

We updated this statement in the cover letter.

5. Thank you for stating the following in the Acknowledgments Section of your manuscript: "ML was supported by a CIFRE PhD Fellowship founded by TRIBUN HEALTH, Paris, France and ANRT (CIFRE 2019/1905). Furthermore, this work was supported by the Ministère de l’Enseignement supérieur, de la Recherche et de l’Innovation under management of Agence Nationale de la Recherche as part of the “Investissements d'avenir” program, reference ANR-19-P3IA-0001 (PRAIRIE 3IA Institute). TW acknowledges financial support from ITMO Cancer (20CM107-00)."

We note that you have provided funding information that is not currently declared in your Funding Statement. However, funding information should not appear in the Acknowledgments section or other areas of your manuscript. We will only publish funding information present in the Funding Statement section of the online submission form. Please remove any funding-related text from the manuscript and let us know how you would like to update your Funding Statement. Currently, your Funding Statement reads as follows: "ML was supported by a CIFRE PhD Fellowship founded by TRIBUN HEALTH, Paris, France and ANRT (CIFRE 2019/1905). Furthermore, this work was supported by the Ministère de l’Enseignement supérieur, de la Recherche et de l’Innovation under management of Agence Nationale de la Recherche as part of the “Investissements d'avenir” program, reference ANR-19-P3IA-0001 (PRAIRIE 3IA Institute). "

Thank you for your comment. Funding-related text was removed from the manuscript. The amended statements were included within the cover letter.

6. We note that you have indicated that there are restrictions to data sharing for this study. For studies involving human research participant data or other sensitive data, we encourage authors to share de-identified or anonymized data. However, when data cannot be publicly shared for ethical reasons, we allow authors to make their data sets available upon request. For information on unacceptable data access restrictions, please see http://journals.plos.org/plosone/s/data-availability#loc-unacceptable-data-access-restrictions. Before we proceed with your manuscript, please address the following prompts:

and

7. When completing the data availability statement of the submission form, you indicated that you will make your data available on acceptance. We strongly recommend all authors decide on a data sharing plan before acceptance, as the process can be lengthy and hold up publication timelines. Please note that, though access restrictions are acceptable now, your entire data will need to be made freely accessible if your manuscript is accepted for publication. This policy applies to all data except where public deposition would breach compliance with the protocol approved by your research ethics board. If you are unable to adhere to our open data policy, please kindly revise your statement to explain your reasoning and we will seek the editor's input on an exemption. Please be assured that, once you have provided your new statement, the assessment of your exemption will not hold up the peer review process.

Thank you for your comment. Due to regulatory requirements of French and European laws concerning the protection of patient data (GDPR (Regulation (EU) 2016/679 of the European Parliament and of the Council of 27 April 2016)), including anonymized anatomical and pathological lesion data, it is not possible to share the dataset openly in accordance with the legal framework. Data sharing between the project members was strictly governed by a contract and approved by the CERAPHP. Centre – Institutional Review Board registration #0001192, which allowed sharing strictly for this specific research project and between the designated parties (APHP, Tribun and CBIO Mines Paristech). In accordance with the regulations, public deposition would breach compliance with the protocol approved by the research ethics board. We propose that access to the dataset could be provided upon request, subject to approval from the Ethics Committee of AP-HP (Assistance Publique - Hôpitaux de Paris). For more information, please contact Dr Anne-Sophie Jannot, Vice-President of the CERAPHP Committee (anne-sophie.jannot@aphp.fr).

The Data Availability statement was updated accordingly in the submission form.

8. Please include your full ethics statement in the ‘Methods’ section of your manuscript file. In your statement, please include the full name of the IRB or ethics committee who approved or waived your study, as well as whether or not you obtained informed written or verbal consent. If consent was waived for your study, please include this information in your statement as well.

Thank you for your remark. Full ethics statement was included in the ‘Methods’ section of the manuscript file (lines 101-104).

9. Please include captions for your Supporting Information files at the end of your manuscript, and update any in-text citations to match accordingly.

Thank you for your comment. Captions for Supporting Information files were added at the end of the manuscript.

We will now turn to a point-to-point answer to the reviewer’s comments.

Reviewer #1:

1. The performance of a single AI model is shown, in a single region of France. How sure are the authors that the problem of overreliance is general and not specific to the setting? What about different AI models?

Thank you for your question. This study focused on a single AI model; however, the panel members were recruited from various hospitals across France, including Paris and the Greater Paris Region (Créteil and Jossigny), as well as Nantes, Brest, Toulouse, and Lille. This diverse panel comprised pathologists from different professional backgrounds, including university hospitals, general hospitals, and comprehensive cancer centers, with varying levels of experience (from recently board-certified to specialists in head and neck pathology). We believe that this approach provides a robust framework for exploring potential overreliance on AI from multiple perspectives.

The AI model utilized in this study is an adaptation of the widely used Multiple Instance Learning framework (Ilse et al.) for image classification. While our research focused on this specific model and task, other AI applications, such as object detection (e.g., immunohistochemistry marking or mitotic figure identification), are also susceptible to overreliance. For instance, the model may correctly highlight objects of interest, but interpreting these detections (determining whether they represent true immunomarkings or actual mitotic figures) can remain challenging in certain cases. Moreover, errors in detection, such as falsely identified objects, might be more readily accepted by pathologists, potentially compounding the issue. Unfortunately, this topic is underexplored in the literature, where most studies emphasize performance improvements rather than addressing the limitations of AI models.

Radiologists, who began using AI models earlier than pathologists, are now starting to address the issue of AI overreliance. However, there is still a limited number of publications exploring this topic in depth (Li MD, Little BP. Appropriate Reliance on Artificial Intelligence in Radiology Education. J Am Coll Radiol. 2023; Akudjedu TN, Torre S, Khine R, Katsifarakis D, Newman D, Malamateniou C. Knowledge, perceptions, and expectations of Artificial intelligence in radiography practice: A global radiography workforce survey. J Med Imaging Radiat Sci. 2023; Rubin DL. Artificial Intelligence in Imaging: The Radiologist's Role. J Am Coll Radiol. 2019).

Finally, we would like to emphasize that the problem of AI overreliance extends beyond medical AI applications. It is a broader issue, as highlighted by Passi and Vorvoreanu in their review of AI overreliance across various fields, including more than 60 papers from different research areas (Passi S, Vorvoreanu M. Overreliance on AI literature review. Microsoft Research, 2022)

2. What is the background on use of the AI model? Is this a model validated and in clinical use in France? Is this model currently in use in diagnostics? You mention how it was trained and developed, but after development, a validation set was used before using the model in clinical practice?

Thank you for your question. This model was developed through a research collaboration between Hôpital Européen Georges Pompidou (AP-HP Paris Hospitals), École des Mines de Paris, and Tribun Health (France), who jointly supervised two theses (MD and PhD) during the project. Our initial results were published in Histopathology (Lubrano M, Bellahsen-Harrar Y, Berlemont S, Atallah S, Vaz E, Walter T, Badoual C. Diagnosis with confidence: deep learning for reliable classification of laryngeal dysplasia. Histopathology, 2023). While the model is not yet implemented in clinical practice, a patent for its confidence score is currently under review.

The model was validated using an external dataset from Hôpital Tenon (AP-HP) and is currently being considered for inclusion in an international validation study. This study aims to confirm the model's robustness before it can be broadly adopted.

3. Is this a model routinely used on a daily basis or a RUO model? Are people using this model also using other AI models? Have they been trained and informed on the use of confidence scores? Are they aware of the potential problems of the model?

Thank you for your question. This model is currently designated as a research-use-only (RUO) model. The individuals participating in this study do not utilize other AI models in clinical practice, as AI implementation in French hospitals remains in its early stages.

Participants were trained to use the model and informed about its potential biases through a tutorial provided prior to the start of their review. This included an online presentation and a written tutorial, which they could access at any time as needed.

4. Please include an experiment on how to avoid these errors. Please show if (or not) training of pathologists on potential errors (before use of the AI) can make a difference. Or maybe a pop-up or warning of low confidence can help. Anything that mitigates these problems.

Thank you for your recommendation. Although we are unable to ask participants to conduct new reviews at this time, we believe our findings demonstrate that improved training reduces AI overreliance, as evidenced by the fact that Head and Neck specialists made fewer model-induced errors.

As shown in S2 Table, carcinomas missed by pathologists during the AI-assisted review but correctly diagnosed during the unassisted review were associated with reviewer 1 (a recently board-certified pathologist) twice, and with reviewers 5 (a community pathologist not specialized in Head and Neck), 6 (a community pathologist specialized in Head and Neck), and 7 (an academic pathologist specialized in Head and Neck) once each. Interestingly, errors made by specialized pathologists had a lesser impact compared to their non-specialized colleagues. For example, reviewer 6 accepted the AI prediction of high-grade dysplasia, which has a less severe outcome than completely missing an invasive lesion or mischaracterizing it as low-grade. Reviewer 7, while unable to make the correct diagnosis, rejected the AI's low-grade dysplasia prediction and chose high-grade dysplasia instead. In contrast, less experienced pathologists (reviewers 1 and 5) followed the AI prediction, which was significantly inaccurate (low-grade dysplasia or non-dysplastic).

To assist pathologists with less experience in this complex field, we stress out the potential benefits of incorporating a confidence score. Our results indicate that a written protocol and oral explanation about model usage are insufficient. This study could serve as a practical tutorial for pathologists, as participants were provided anonymous access to their results afterward. Implement

---

## [Decision Letter · Decision Letter 1]

7 Mar 2025

PONE-D-24-53739R1AI in diagnostic pathology: exploring the risks of over-reliance and its clinical consequences. What lessons can be learned to support the training of young pathologists?PLOS ONE

Dear Dr. Bellahsen-Harrar,

Thank you for submitting your manuscript to PLOS ONE. After careful consideration, we feel that it has merit but does not fully meet PLOS ONE’s publication criteria as it currently stands. Therefore, we invite you to submit a revised version of the manuscript that addresses the points raised during the review process.

We look forward to receiving your revised manuscript.

Kind regards,

Mohammad Amin Fraiwan

Academic Editor

PLOS ONE

Journal Requirements:

Reviewers' comments:

Reviewer's Responses to Questions

**Comments to the Author**

1. If the authors have adequately addressed your comments raised in a previous round of review and you feel that this manuscript is now acceptable for publication, you may indicate that here to bypass the “Comments to the Author” section, enter your conflict of interest statement in the “Confidential to Editor” section, and submit your "Accept" recommendation.

Reviewer #1: (No Response)

2. Is the manuscript technically sound, and do the data support the conclusions?

Reviewer #1: Yes

3. Has the statistical analysis been performed appropriately and rigorously? 

Reviewer #1: I Don't Know

4. Have the authors made all data underlying the findings in their manuscript fully available?

Reviewer #1: Yes

5. Is the manuscript presented in an intelligible fashion and written in standard English?

Reviewer #1: Yes

6. Review Comments to the Author

Reviewer #1: I still think the authors should focus still more on addressing how to avoid the errors. There is no proof provided whatsoever that shows the effect of training to mitigate the problem. I think this is a flaw. It should be addressed, at least in the text, and at least how this kind of training should be done.

7. PLOS authors have the option to publish the peer review history of their article (what does this mean? ). If published, this will include your full peer review and any attached files.

**Do you want your identity to be public for this peer review?** For information about this choice, including consent withdrawal, please see our Privacy Policy .

Reviewer #1: No

---

## [Author Response · Author response to Decision Letter 2]

3 Apr 2025

Reviewer #1 :

Thank you for your remark. In our study, we did not directly demonstrate that training younger and non-specialized pathologists on the panel would enhance their ability to use AI effectively and improve diagnostic accuracy. To establish this, a repeated evaluation of the slides by pathologists would be necessary to observe whether their performance improves through iterative learning from errors. However, our dataset is limited, particularly concerning misinterpreted cases. This raises the concern that pathologists may recall their previous incorrect diagnoses and provide correct answers through memory recall rather than genuine improvement in diagnostic skills, in the same way an AI model can be overfit on limited datasets. Such investigations would be more appropriately conducted in a longitudinal study, tracking pathologists from the initial stages of their training through their early years of board-certified practice.

In radiology, a field with substantial AI integration in daily practice, previous studies have indicated that younger radiologists are particularly prone to automation bias, resulting in an over-reliance on AI predictions (1), (2), (3). A study by Mehrizi et al. (4) suggested that even increased explainability does not mitigate this issue. Our results suggest the same problem applies to pathology. Specifically, in cases where AI led to incorrect diagnoses despite accurate unassisted reviews, the more experienced pathologists considered the AI's input but did not follow it uncritically. In contrast, recently board-certified and non-specialized pathologists were more likely to rely on the AI's suggestions, which may show that this tendency to automated bias in younger practitioners might be related to the “figure of authority” described in cognitive development science, here consisting in a mathematical algorithm, which operating mode can seem not understandable, and thus not really questionable, when lacking a strong opinion and confidence from the beginning (5). These considerations underscore the need for alternative approaches to assist pathologists in using AI effectively.

To address these challenges, radiology societies in North America have introduced structured educational programs, such as the National Imaging Informatics Course – Radiology and the Radiological Society of North America Imaging AI Certificate (6), (7). These courses teach how AI models work and their ability to induce biased judgments. Pathology societies are starting to follow a similar trajectory, offering AI-focused training that emphasizes the recognition of AI limitations and the risks associated with bias (8). In France, the integration of a dedicated AI mandatory course is quite recent (2023), and it has yet to include a specific course on AI biases.

In our opinion, the confidence score can be of great use to raise awareness on AI limitations on specific cases. Evaluating its influence on pathologist training outcomes should require a dedicated, long-term study.

These elements were integrated to the revised manuscript (text in green – lines 344-386).

---

## [Editor Report · Decision Letter 2]

6 Apr 2025

AI in diagnostic pathology: exploring the risks of over-reliance and its clinical consequences. What lessons can be learned to support the training of young pathologists?

PONE-D-24-53739R2

Dear Dr. Bellahsen-Harrar,

We’re pleased to inform you that your manuscript has been judged scientifically suitable for publication and will be formally accepted for publication once it meets all outstanding technical requirements.

Kind regards,

Mohammad Amin Fraiwan

Academic Editor

PLOS ONE
---

## [Editor Report · Acceptance letter]

PONE-D-24-53739R2

PLOS ONE

Dear Dr. Bellahsen-Harrar,

I'm pleased to inform you that your manuscript has been deemed suitable for publication in PLOS ONE. Congratulations! Your manuscript is now being handed over to our production team.

Kind regards,

on behalf of

Dr. Mohammad Amin Fraiwan

Academic Editor

PLOS ONE